# A Toxicity Screening Approach to Identify Bacteriophage-Encoded Anti-Microbial Proteins

**DOI:** 10.3390/v11111057

**Published:** 2019-11-14

**Authors:** Ushanandini Mohanraj, Xing Wan, Cindy M. Spruit, Mikael Skurnik, Maria I. Pajunen

**Affiliations:** 1Department of Bacteriology and Immunology, Medicum, Human Microbiome Research Program, Faculty of Medicine, University of Helsinki, 00290 Helsinki, Finland; ushanandini.mohanraj@helsinki.fi (U.M.); xing.wan@helsinki.fi (X.W.); c.m.spruit@uu.nl (C.M.S.); mikael.skurnik@helsinki.fi (M.S.); 2Department of Virology, Medicum, University of Helsinki, 00290 Helsinki, Finland; 3Division Animal and Human Health Engineering, Kasteelpark Arenberg 21 - box 2462, 3001 Leuven, Belgium; 4Laboratory of Microbiology, Wageningen University and Research, 6708 WE Wageningen, The Netherlands; 5Division of Clinical Microbiology, Helsinki University Hospital, HUSLAB, 00290 Helsinki, Finland

**Keywords:** HPUF, antibacterials, screening, bacteriophages, assay, toxic, φR1-RT

## Abstract

The rapid emergence of antibiotic resistance among many pathogenic bacteria has created a profound need to discover new alternatives to antibiotics. Bacteriophages, the viruses of microbes, express special proteins to overtake the metabolism of the bacterial host they infect, the best known of which are involved in bacterial lysis. However, the functions of majority of bacteriophage encoded gene products are not known, i.e., they represent the hypothetical proteins of unknown function (HPUFs). In the current study we present a phage genomics-based screening approach to identify phage HPUFs with antibacterial activity with a long-term goal to use them as leads to find unknown targets to develop novel antibacterial compounds. The screening assay is based on the inhibition of bacterial growth when a toxic gene is expression-cloned into a plasmid vector. It utilizes an optimized plating assay producing a significant difference in the number of transformants after ligation of the toxic and non-toxic genes into a cloning vector. The screening assay was first tested and optimized using several known toxic and non-toxic genes. Then, it was applied to screen 94 HPUFs of bacteriophage φR1-RT, and identified four HPUFs that were toxic to *Escherichia coli*. This optimized assay is in principle useful in the search for bactericidal proteins of any phage, and also opens new possibilities to understanding the strategies bacteriophages use to overtake bacterial hosts.

## 1. Introduction

The widespread appearance of drug-resistant bacterial isolates has a profound burden on health care [1]. Significant efforts have been made for the development of new and effective antibacterials to meet this challenge. A variety of natural compounds from different organisms have been found with antimicrobial activity [2]. When compared with natural product sources such as plants and animals, microorganisms are much more readily renewable and reproducible as a source. In recent years, viruses, especially bacteriophages, have emerged as a very potent source of antimicrobials [3].

Bacteriophages are viruses that infect bacterial cells [4]. Recently, there have been attempts to develop antibacterial agents from the bacteriostatic and bacteriolytic proteins encoded by bacteriophages, e.g., phage-encoded endolysins, VAPGH (virion-associated peptidoglycan hydrolases), polysaccharide depolymerases, and holins [3]. Bacterial cell lysis is a prerequisite for phage progeny release and dispersion for infection of new hosts. The majority of the phages lyse their bacterial hosts with the help of lytic enzymes by compromising the structural integrity of the peptidoglycan layer [5]. Hence, among the most promising alternatives to conventional antibiotics are phage-derived lytic enzymes.

Apart from these known lytic proteins, phages encode other gene products for which functions are not known. Such proteins of unknown function are called “hypothetical proteins of unknown function” (HPUF) [6]. Many hypothetical phage proteins are short polypeptides, and some are shown to inhibit the growth of the bacterial host [7,8,9]. There is an accumulation of a large number of proteins of unknown function in the databases due to the increasing gap between predicted phage gene sequences and their corresponding functions [10]. From what is already known, given their abundance and diversity, phage genomes could be packed with new genes involved in host inhibition. These would provide a rich source of leads for the development of novel antibacterials.

In the current study we describe a phage genomics-based screening approach of HPUFs to identify phage-encoded toxic products. Using this approach, we screened 94 HPUFs of *Yersinia* phage φR1-RT and identified four HPUFs toxic to *Escherichia coli*. Bacteriophage φR1-RT is distantly related to phage T4, and infects a number of Gram-negative *Yersinia enterocolitica* strains [11]. *Y. enterocolitica*, a member of the *Enterobacteriaceae* family, is a zoonotic pathogen causing human and animal enteric infections, including acute diarrhea, enterocolitis and terminal ileitis, and in rare cases sequelae such as reactive arthritis and erythema nodosum. When spread systemically, for example due to contaminated products used in blood transfusion, it can also cause fatal sepsis [12]. In the current work we demonstrated that using our screening approach, toxic proteins could be systematically identified among phage-encoded HPUFs. We further suggested that the toxic proteins in turn could be used to identify new bacterial targets for drug discovery.

## 2. Materials and Methods

### 2.1. Bacterial Strains, Plasmids, Bacteriophages and Culture Conditions

Bacterial strains, bacteriophages and plasmids used in this work are listed in Table 1. *E. coli* laboratory strain DH10B and DH5α were used for cloning. Electrocompetent *E. coli* cells were prepared essentially as described previously [13]. LB agar plates were solidified with 1.5% (*w*/*v*) of agar. M9t minimal medium contained 3.4 mM Na_2_HPO_4_, 2.2 mM KH_2_PO_4_, 0.86 mM NaCl, 0.94 mM NH_4_Cl, 0.2% (*w*/*v*) tryptone, 2.0 mM MgSO_4_, 0.10 mM CaCl_2_ and 3.0×10^−3^ mM vitamin B1 (Merck KGaA, Darmstadt, Germany). Media (agar, tryptone and yeast extract) were purchased from Neogen Food Safety, Lansing, MI, USA. When required, 100 µg/mL ampicillin (Amp; Sigma-Aldrich, St. Louis, MO, USA) was used.

### 2.2. Mass Spectrometry (MS) Data Analysis of Phage Proteins

The LC-MS/MS experiments were carried out at the Proteomics Unit, Institute of Biotechnology, University of Helsinki, using the protocols described elsewhere [17]. The obtained peptide sequences from the MS were compared simultaneously against the phage and bacterial (*Y. enterocolitica* serotype O:3 strain Y11 – Accession number FR729477.2) protein sequences. From the LC-MS/MS analysis, proteins which fulfilled the inclusion criteria (>2 unique peptides and/or >5% coverage) were considered for annotation as phage particle associated proteins. The hypothetical proteins that were identified in the LC-MS/MS data were re-annotated to hypothetical structural proteins in the φR1-RT genome sequence under the accession number HE956709.

### 2.3. Comparative Protein Analysis of Hypothetical Proteins of Unknown Function (HPUFs) and Genome Annotation

To re-evaluate the putative functions of the HPUFs, similarity searches were performed using the NCBI BLASTp tool [18] against the non-redundant protein database to confirm the absence of any HPUF homologs with known functions. Proteins with low query coverage (<50%) or low sequence identity (<20%) were not considered, while proteins with high sequence identities (>40%), and low *E*-value (<0.005) were considered as close homologs. In addition to BLASTp, Hidden Markov Models (HMMs)-based HHpred tool [19] was used, which provided domain-based similarity searching for the query sequence. HHpred tool search was performed against a database of PDB, SCOP and Pfam to identify the putative domains and functions of individual HPUFs.

### 2.4. PCR Amplification of Control and Hypothetical Proteins of Unknown Function Genes for Cloning

The sequences of the genes amplified in the current study were retrieved from GenBank. All the primers were designed manually according to [20] and sent to Metabion (Germany) for synthesis. Appendix A list the sequences of all the primers that were designed and used in the present study.

Reactions were performed either in 0.2 mL thin walled PCR tubes (*4titude*^®^ Ltd., Wotton, UK), in 50 μL volumes containing 10 ng of DNA template for single tube reactions, or in 96 well microtiter plate format using 30 μL volumes containing 5 ng of DNA template. For each reaction, 0.5 μM of primers, 0.2 mM of dNTP mix (Thermo Fisher Scientific, Waltham, MA, USA), 10 or 6 μL of 5× Phusion Buffer and 0.02 U/μL Phusion DNA Polymerase (Thermo Fisher Scientific, Waltham, MA, USA) were added. The PCR cycling included an initial denaturation at 98 °C for 30 s followed by 30 cycles, each consisting of 10 s at 98 °C, 30 s at Tm and 30 s at 72 °C for extension, followed by a final extension step at 72 °C for 10 min. The Tm, annealing temperature, was calculated for each primer pair using the web-tool at: https://www.thermofisher.com/in/en/home/brands/thermo-scientific/molecular-biology/molecular-biology-learning-center/molecular-biology-resource-library/thermo-scientific-web-tools/tm-calculator.html. The PCR products were kept on hold, at 4 °C, until further processing. When using the phage particles directly as a template for PCR, 2 μL (~60 ng) of phage particle suspension was used as a template with initial denaturation at 98 °C for 3 min. The rest of the PCR cycle steps were as described above.

### 2.5. Cloning

The toxic and non-toxic control genes, and the φR1-RT HPUFs genes were cloned into the multiple cloning site of the pUC19-based vector pU11L4 using the *Not*I and *Nco*I restriction sites (Thermo Fisher Scientific, Waltham, MA, USA) (Appendix A). Whenever an internal *Nco*I site was present in any of the HPUF-genes, the *Nhe*I (Thermo Fisher Scientific, Waltham, MA, USA) digestion was used instead.

All the ligation reactions in the present study were prepared in final volumes of 15 μl consisting of 75 ng of vector with a vector to insert ratio 1:3, 1.5 μl of 10X T4 Ligation Buffer (NEB, Ipswich, MA, USA) and 3 U of T4 DNA Ligase (NEB, Ipswich, MA, USA). The reactions were incubated overnight at 16 °C, heat inactivated at 65 °C for 20 min and stored at −20 °C until further use.

Transformations were done by electroporation with the exponential decay pulses generated by a Gene Pulser™ apparatus (Bio-Rad Laboratories, Hercules, CA, USA) set at 25 μF and 2.5 kV for 2 mm cuvettes. The competent cells were first thawed on ice. Then 45 μL of cells and 2 μL of ligated DNA solution were transferred to a cold, 1.5 mL polypropylene tube; the suspension was mixed by flicking the tube. The cell/DNA mixture was then added to the chilled cuvettes, and the appropriate pulse applied. Following the pulse, the cells were immediately removed from the electrodes and mixed into 950 μL of SOC outgrowth medium (SOC: 2% Bacto tryptone, 0.5% Bacto yeast extract, 10 mM NaCl, 2.5mM KCI, 10 mM MgCl₂, 10 mM MgSO_4_, 20 mM glucose). The samples were incubated, with shaking at 225 rpm, for 1 h at 37 °C. After this, the cells were plated on LB-agar containing selective antibiotics to screen for transformants.

### 2.6. Screening for Toxic Hypothetical Proteins of Unknown Function Genes

All the 94 hypothetical genes were PCR amplified, restricted and ligated as previously described. After electroporation, the transformation mix was plated on the LB-Ampicillin agar plates. The plates were kept at 37 °C overnight and next day colonies were counted using a colony counter. Each screening experiment included the HPUF genes and at least one non-toxic control gene. Colony forming units (CFU) were normalized to the CFU of the non-toxic control gene in an individual experiment and expressed as relative CFUs.

### 2.7. Confirmation of Protein Toxicity

Potentially toxic φR1-RT HPUFs that were selected in the initial screening assay and control genes *g150* and *lysE* were cloned into the pBAD30 vector [16] using *Eco*RI and *Xba*I (Thermo Fisher Scientific, Waltham, MA, USA) restriction sites. Whenever an internal *Xba*I restriction site was present, cloning was performed with *Pae*I, whereas *EcoR*I was replaced with *Kpn*I (Thermo Fisher Scientific, Waltham, MA, USA) instead. The plasmids were purified and the presence of the correct inserts was confirmed by sequencing at the Institute for Molecular Medicine Finland Technology Centre Sequencing Unit [21]. The plasmids were electroporated into electrocompetent *E. coli* DH5α. Three single colonies per gene were inoculated into 1 mL LB medium supplemented with 0.2% (*w*/*v*) glucose and 100 µg/mL ampicillin (Amp; Sigma-Aldrich, St. Louis, MO, USA) in a 2 mL Eppendorf tube and incubated overnight at 37 °C with shaking at 160 rpm. The following day, the bacteria were collected by centrifuging for 5 min at 8000 rcf and the medium was replaced with 1 mL minimal medium M9t. Afterwards, M9t medium was supplemented with 100 µg/mL Amp and either 0.2% (*w*/*v*) glucose or 0.2% (*w*/*v*) arabinose was inoculated with 1% inoculum of washed bacterial cells. The bacterial dilutions were transferred to Bioscreen Honeycomb 2 plates in triplicate (300 µL/well). The OD at 600 nm was measured every hour for 16 h using the Bioscreen C MBR (Oy Growth Curves Ab Ltd., Helsinki, Finland). The plate was shaken continuously with high amplitude and normal speed. Shaking was stopped 10 s before measuring and measuring was started immediately. Average values over the triplicate measurements were taken. The overall average values and standard deviations were calculated over the three biological replicates.

### 2.8. Structure Prediction of Toxic HPUFs

Modeling and secondary structure prediction of the hypothetical proteins was performed using the online Phyre2 modelling server accessed on 15.07.2019 [22].

### 2.9. Statistics

Graphical representations and statistical analysis were performed using Origin 7.5 Software (OriginLab Corp., Northampton, MA, USA) with data presented as the means ± standard deviation (SD).

## 3. Results

### 3.1. Elimination of Hypothetical Proteins of Unknown Function (HPUFs) by LC-MS/MS Analysis of Phage Particle Proteomes

In the φR1-RT genome (HE956709), 129 predicted genes were annotated to encode HPUFs. To exclude the structural genes from the screening approach the phage particle structural proteins of the φR1-RT phage were identified using the LC-MS/MS proteomics data. Altogether, 78 phage proteins that fulfilled the inclusion criteria were identified in the LC-MS/MS analysis (Appendix A). These included 21 proteins earlier annotated as HPUFs, and based on the LC-MS/MS data they were re-annotated as putative structural proteins and eliminated from the toxic protein screening approach (Appendix A).

### 3.2. Further Elimination of Hypothetical Proteins of Unknown Function (HPUFs) by BLASTp and HHpred Analyses

The remaining 108 HPUFs were subjected to BLASTp and HHpred searches to identify other HPUFs with predictable functions to be eliminated from the screening (Appendix A).

Of the 108 hypothetical proteins, Gp019, Gp138, and Gp255, had no significant hits in the BLASTp search. On the other hand, nine were found to show significant similarity to proteins of known function; of these, six were involved in nucleotide metabolism, and three as phage structural proteins. All the others had hypothetical proteins as major BLASTp hits. Putative conserved domains were detected for 21 proteins. Out of the 21, 7 had domains that belonged to the protein families of unknown function. A transmembrane helix was detected for Gp076.

In the HHpred search only 39 proteins had significant hits. One transmembrane helix was detected in Gp076 and Gp094, and two transmembrane helices were detected in Gp105 and Gp137. In addition, coiled-coil segments were detected in Gp087, Gp098, and Gp143.

In summary, based on the BLASTp and HHpred searches, altogether 14 HPUFs were further removed from the true HPUFs; eight (Gp139, Gp143, Gp183, Gp184, Gp185, Gp251, Gp038, and Gp213) that were similar to other known structural proteins, five (Gp039, Gp047, Gp057, Gp075, and Gp179) that had domains similar to proteins required for DNA replication, transcription, translation or nucleotide metabolism, and one (Gp107) that was similar to rI lysis inhibition regulator membrane protein of *Yersinia* phage vB_YenM_TG1. Hence, the remaining 94 HPUFs were selected to be screened for toxic ones.

### 3.3. Ligations of Toxic and Non-Toxic Control Genes Show Differences in Plating Efficiency

Based on the obtained proteomics data the *g121*, *g178*, *g246* and *g150* genes of φR1-RT were selected as non-toxic controls, and based on a literature search, the *ndd*, *dcd-1l*, *lysE* and *regB* genes were selected as toxic controls. When the *E. coli* DH10B cells were transformed with mixtures containing plasmid pU11L4 ligated with PCR fragments of the control genes encoding toxic and non-toxic proteins, significant differences were observed between the number of transformants obtained (Figure 1). The *regB* and *dcd-1l* genes were observed to be more toxic to *E. coli* cells as compared to *lysE* and *ndd*. We also observed that the non-toxic control gene *g121* resulted in lower transformation efficiency as compared to the other non-toxic control genes. Based on the difference in the transformation efficiencies observed between the different control toxic and non-toxic genes, an assay was developed to screen for toxic HPUF genes.

### 3.4. Eight Potential Toxic Protein Hits Identified from an Initial CFU-Based Screening Approach

After successful amplification of all the 94 HPUF genes, they were digested for ligation by appropriate restriction digestions, and ligated with the pU11L4 plasmid. The obtained ligation mixtures were transformed into *E. coli* DH10B cells. CFUs were counted for all the 94 HPUFs and relative CFU values were calculated (Appendix A). Figure 2A–C represents relative CFU values for all the 94 HPUF genes normalized to the non-toxic control gene included in each individual transformation experiment. Based on the observation of the CFU values of the control toxic genes and their difference with the non-toxic genes, a cut-off value of 0.4 (40% of the non-toxic control) was chosen for the screening by plating assay. HPUFs displaying a relative CFU value equal or lower than 0.4 upon transformation were considered toxic and further analyzed (Figure 2A,B). When considering *g121* as a non-toxic control, instead of considering hits below a relative CFU value of 0.4, hits below relative CFU 0.5 were considered since *g121* gave consistently lower CFU values as compared to the other non-toxic controls (Figure 2C). From this initial screening, eight potential hits toxic to *E. coli* cells were identified.

### 3.5. Toxic Protein Hits Gp064, Gp136, Gp137 and Gp232 Inhibit the Growth of E. coli Cells

To confirm the toxic hits obtained in the initial screening, the genes of the eight potentially toxic HPUFs were cloned into pBAD30 under the control of the arabinose promoter P_BAD_. In the presence of arabinose, the *E. coli* DH5α bacteria carrying the plasmids will express the HPUF gene, whereas in the presence of glucose, the expression is repressed. To avoid the possible influence of glucose present in a rich LB medium, a semi-defined minimal medium M9t was used. The recombinant bacteria with *g150*, the known non-toxic control gene, showed similar growth both under inducing and repressing conditions. This was in contrast to the recombinants expressing the toxic control gene, *lysE*, or four of the eight toxic HPUF hits, i.e., *g064*, *g136*, *g137* and *g232.* These showed retarded growth in the presence of arabinose (Figure 3) and were thus regarded as toxic when expressed in *E. coli*.

### 3.6. Modelling Studies

The Phyre2 software was applied to model putative functional domains and/or secondary structures for the toxic HPUFs Gp064, Gp136, Gp137 and Gp232. Except for Gp232, the models produced had very low confidence levels. Interestingly, for Gp232, a stretch of 92 residues could be reliably modelled with 98.7% confidence and 32% identity using the homing endonuclease (PDB ID: 3R3P) from *Bacillus* phage 0305phi8-36 as the template (Figure 4). In Phyre2, the query and the template sequence identity should be at least 30–40% and the confidence level > 90%, for a reliable model [22]. Additionally, only one transmembrane helix was predicted for Gp136, and two transmembrane helices for Gp137 (Figure 5).

## 4. Discussion

With the accumulation of sequence data on thousands of new bacteriophage genomes annually we still face the fact that a majority of the predicted genes encode proteins (i.e., HPUFs) either with no similarity to any proteins in the databases, or similarity to proteins with unknown functions [23,24]. Hence, toxicity screening of these HPUFs, and subsequent identification of their mode of action will likely reveal novel targets for antibacterial agents. It is unlikely that the expression of any toxic phage product would be continuous throughout the phage infection cycle, instead it likely occurs at a certain phase of the infection [25]. In the current study, a phage genomics-based screening method was designed, and as a proof of principle, the HPUFs of phage φR1-RT were screened for toxic ones. The phage φR1-RT genome (HE956709) was 168 kb in size with 262 predicted genes [11]. Based on similarity searches, putative functions were assigned to only 133 gene products (50.7%) of phage φR1-RT, thus the remaining 129 gene products were annotated as HPUFs.

A proteomic analysis carried out on purified phage particles has allowed for the assignment of some of the HPUFs as structural proteins [26]. From an antibacterial point of view, our interest lies in the phage proteins that would inhibit the host bacteria upon expression, and hence the structural proteins would not belong to this group of HPUFs. Proteins that are not identified in the LC-MS/MS analysis are likely the proteins whose expression takes place in the host and are not part of the virion. Hence LC-MS/MS analysis of purified phage particles directly measures peptides arising from the structural phage proteins. However, there is a possibility that some of the identified proteins are carried over from the lysate. A phage lysate is obtained by infecting host bacteria with the phage and letting the phage lyse the cells [27]. It cannot be excluded that some bacterial proteins will be co-isolated with the phage particles and identified due to the high sensitivity of the method. Hence the obtained peptide sequences from the LC-MS/MS were compared simultaneously against not only the phage but also the host bacterial (*Y. enterocolitica* serotype O:3 strain Y11 - FR729477.2) protein sequences as well. On the other hand, some phage structural proteins might not be detected due to their low abundance in the phage particles. Some of these were identified using BLASTp and/or HHpred analysis. HHpred is known for its sensitivity in detecting remote protein homology, for structure predictions, and it uses a pairwise comparison of profile HMMs [19].

From the initial LC-MS/MS data (Appendix A), it was observed that not all of the phage particle-associated proteins were structural proteins e.g., Gp235 was identified as thymidylate synthase, Gp055, as DNA polymerase, Gp003, as DenB, the DNA endonuclease IV, Gp233, as ribonucleotide reductase of class Ia, Gp031, as dCTP pyrophosphatase, and Gp082, as thioredoxin. While it is possible that these proteins are truly associated with the phage particles, they may also possess physical properties that make them more prone to being co-isolated with phage particles during the purification process, or that they might display non-specific binding to the capsid proteins. Some more complex phages package their RNA polymerase into the phage head and eject it into the host cell along with the phage DNA in order to facilitate the transcription of early phage genes [28], supporting the true association. This makes sense as upon infection the first step in the phage life cycle is transcription requiring DNA-dependent RNA polymerase and its associated factors [29]. Furthermore, the bacteriophage T4 gene 42 encodes the dCMP hydroxymethylase, an enzyme unique to the deoxyribonucleotide metabolism of T-even bacteriophages [30]. After eliminating all putative structural HPUFs and those required for transcription or nucleotide metabolism with MS analysis and BLASTp and HHpred, 94 HPUF genes were considered for screening.

In one of the earlier studies involving screening of phage genomes for *S. aureus* growth inhibitory gene products, phage genes were cloned under the control of an arsenite-inducible promoter [7]. Using this method, 31 novel toxic protein families were found from these phage genomes. In their approach, they screened all the predicted phage genes and expression cloned each one of them for toxic protein identification. Another screening approach that has been previously reported for the identification of inhibitory gene products involved the construction of genomic libraries of phages and the identification of inhibitory genes [8,31]. In both approaches, there was a possibility for the presence of structural genes and genes that were involved in the nucleotide metabolism, in the screening process. As these gene products are likely non-toxic their inclusion in the screen would be a waste of resources and effort.

The incomplete repression of promoter combined with the effects of high-copy-number plasmids and transcriptional read-through, would result in the expression of a toxic protein and the killing of the bacterial host [32]. In line with this, we observed that ligation of the toxic genes *regB*, *ndd*, *lysE* and *dcd-1l* into pU11L4 plasmid resulted in a reduced number of transformants as compared to the non-toxic genes *g150*, *g178*, g246 and g121. Based on this we designed our screening approach such that we cloned the ninety-four HPUF genes into pU11L4 plasmids and screened for those that resulted in lower transformation efficiencies as compared to the non-toxic control genes. Furthermore, we confirmed the toxicities of the genes from the initial screening through an arabinose inducible expression system. Our confirmatory experiments on eight HPUF genes suggested a true-positive predictive rate of 50%, which meant that half of the genes predicted to be toxic based on initial screening information were found to inhibit the growth of bacteria under a controlled expression.

From the inter-assay variability experiment (Appendix A), we observed that the coefficient of variance for observed transformation efficiencies from two separate batches of ligation for genes *g033*, *g064*, *g092*, *g100* and *g232* was more that 20%. From the pBAD assay, three of these genes *g033*, *g092* and *g100* were not truly toxic. Hence, by reducing the assay-to-assay variations, the true positive prediction rate could be improved in our screening approach. The variation in the plating efficiency of a single gene could be attributed to experiment-to-experiment variation between setting up the ligation mixtures, the electroporation and plating efficiencies. In our screening approach (Figure 1), we calculated the transformation efficiencies per nanogram of ligation mixture as opposed to the use of purified plasmids. Though use of the ligation mix directly in the screening process saves time, there is the possibility of the ligation mixtures containing widely different absolute numbers of successfully ligated plasmids with inserts to be electroporated into the bacteria. Despite the use of the same ligation conditions and insert to vector ratio for all the experiments, the ligation efficiencies could vary, resulting in different amounts of linear and circular plasmids. Furthermore, variation could also arise during the spreading and growth of the bacteria on the plates, resulting in a different amount of CFUs per plate for the same volume of transformed cells. In future experiments, therefore, we will use ligation and plating replicates in the initial screening assay to minimize variation and reduce false positive hits.

The genes encoding endolysin (*g122*) and holin (*g253*) were previously identified from the φR1-RT genome [11]. Additionally, the gene *g252* product showed similarity to a holin. As the endolysins and holins of many phages have demonstrated antibacterial activity [3], we also tested the φR1-RT endolysin and holin encoding genes in the screening by plating assay for their toxicity towards *E. coli*. To our surprise we did not see the expected reduction in the plating efficiency with any of the genes (Appendix A). A plausible explanation for this is that T4-like endolysins and holins are not active alone, as is the case in the closely related T4 [33]: “In the absence of either the lysozyme or the holin, lysis does not occur”. T4 holin (the gene *t* product) is unlike many other holins not immediately toxic but requires its own endolysin for cell lysis to occur. Analogously, a lysis gene cassette of *Burkholderia pseudomallei* bacteriophage ST79 contains four genes, and minimally for lysis to occur, the expression of the peptidase and holin encoding genes is required, and only moderate lysis was observed when holin alone was expressed [34]. Similar dependence of a phage holin on cognate endolysin was observed for *Streptococcus suis* phage SMP, where the holin alone caused only weak lysis but with endolysin strong lysis [35]. In our screening approach we did not detect these interaction-dependent toxic phage products including proteins that require other phage-encoded proteins for proper folding or for other forms of post-translational modification.

Based on Phyre2 modelling, an endonuclease domain was predicted for Gp232. Endonucleases and other nucleic acid cleaving enzymes belong to a large and extremely diverse family of proteins displaying little sequence similarity though they retain a common core fold responsible for cleavage. Some endonucleases include different types of Zn-binding and DNA-binding domains [36]. The confidence levels of models of all the other toxic hits were low and hence were not reliable. Therefore, our current data does not indicate any putative function for these hits and hence further biochemical studies are required. On the other hand, this may indicate that these hits might constitute a completely novel class of antibacterial proteins or peptides without homology to any currently known protein.

In a previous study [9], affinity purification mass spectrometry was used to identify phage proteins binding to specific bacterial targets. The bacterial targets were selected based on the hypothesis that phages could affect important metabolic proteins, involved in general transcriptional regulation, post-transcriptional regulation, fatty acid biogenesis, cell division, and energy household. Furthermore, many phages are known to encode proteins that interact with the RNA and DNA polymerases of their hosts to either inhibit or redirect bacterial transcription and genome replication. Based on this approach eight proteins that are toxic to *Pseudomonas aeruginosa* were identified, of which four were also toxic to *E. coli*. Though this method is efficient in identifying toxic proteins along with their bacterial targets, this approach limits the identification of toxic proteins interacting with novel bacterial targets.

The results from our study are aimed at forming the basis for phage-host interaction studies, which has a great potential to reveal novel antibacterial mechanisms and molecular targets to treat bacterial infections. Our current fast screening approach for new toxic proteins should be generally applicable to phages infecting relatively closely related bacterial hosts, such as *Enterobacteriaceae* species. The toxic phage proteins are not likely to act as such to kill bacteria extracellularly as they cannot penetrate the cell walls and membranes. Therefore, investigations on the molecular level interactions between the toxic hits and the host bacteria are required to expose their targets, which in turn could be used to screen small molecule libraries for inhibitors. We can think of two possible approaches to identify the targets. First, the phage toxin should be purified to be used as a bait in fishing out the target. As the expression of the protein in *E. coli*, due to its toxicity, may be difficult, yeast or plant cells could be used instead, or ultimately, in vitro translation could be applied. Second, a genetic approach could be used where spontaneous toxin-insensitive mutants would be isolated and the target identification would be carried out by whole genome sequencing of the mutants.

Although only four toxic hits were identified among the 94 φR1-RT HPUFs, we are confident that the genomics-based screening of toxic phage proteins, to be followed by identification of the respective phage-host interactions, as well as structural studies of individual interacting partners has the potential to generate big data for the contemporary drug industry.

## 5. Conclusions

Although it has been stated that functional elucidation of antibacterial phage proteins could be a powerful tool in target identification and drug discovery, only limited progress has been made in the last decade. Our study presents a screening approach to identify toxic phage HPUFs based on their ability to inhibit bacterial growth upon intracellular expression in the host bacteria. In the current study, four toxic HPUFs against *E. coli* were identified. The screening assay allows us to systematically examine and identify toxic genes from hypothetical phage proteins that, as far as we know, are not currently pursued by the industry as targets for antibiotic development.

## Figures and Tables

**Figure 1 viruses-11-01057-f001:**
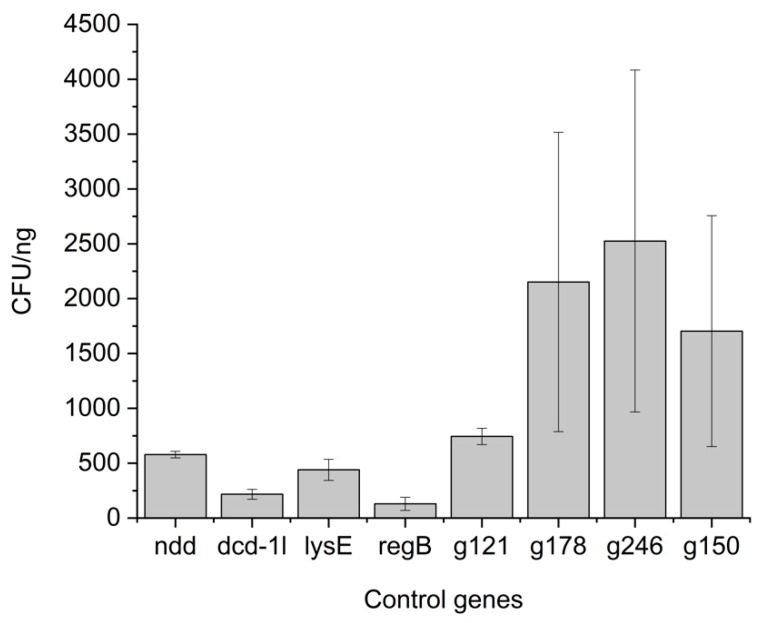
The differences in transformation efficiencies of ligation mixtures of known toxic and non-toxic genes. The bars indicate mean ± *SD* in the CFU/ng values of two different replicates from two different electroporations for each sample.

**Figure 2 viruses-11-01057-f002:**
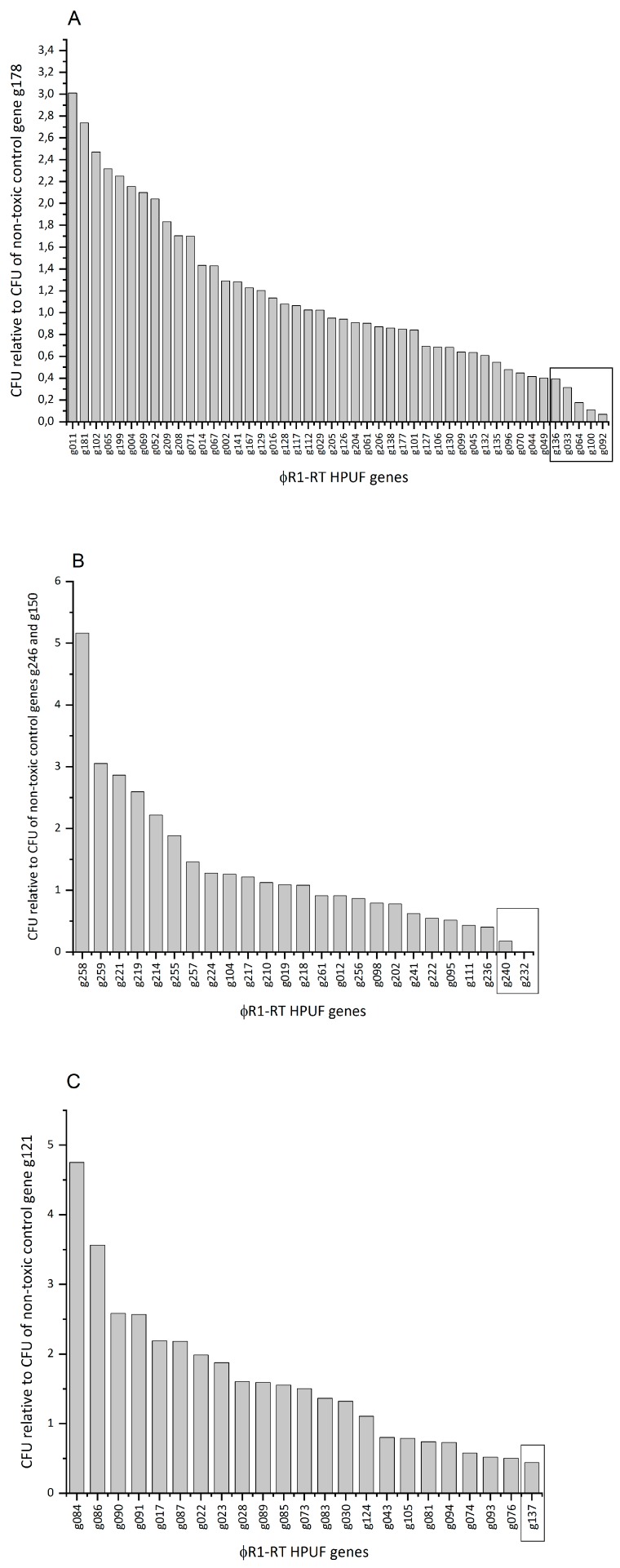
Relative CFU results of the 94 HPUF genes from the screening by plating assay. The plating efficiencies were normalized relative to the control genes *g178* (**A**), *g246* and *g150* (**B**) or *g121* (**C**).

**Figure 3 viruses-11-01057-f003:**
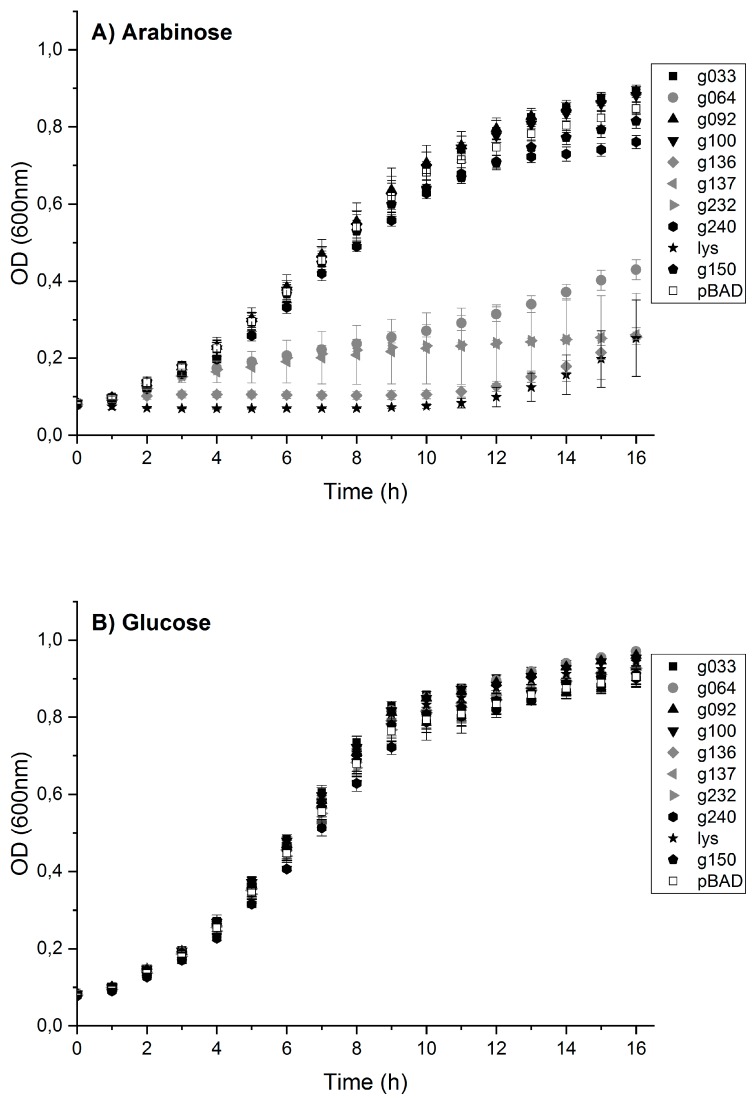
Growth curves of recombinant *E. coli* expressing phage proteins under the control of arabinose inducible promoter in pBAD30 in minimal media M9t supplemented with (**A**) arabinose for gene expression or (**B**) glucose for gene repression.

**Figure 4 viruses-11-01057-f004:**
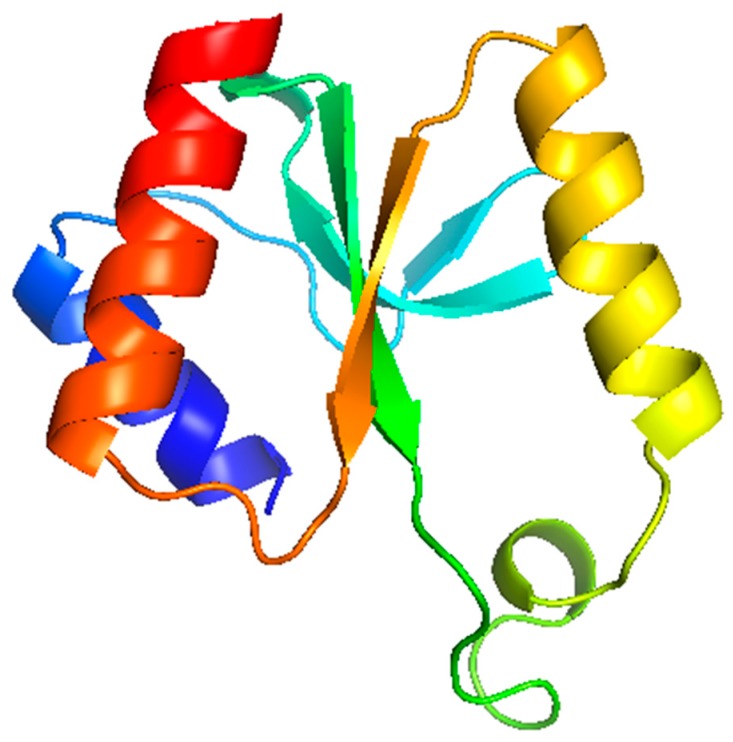
The predicted tertiary structure for the 92 residues (245 aa–344 aa) of Gp232. The image is colored by rainbow N → C terminus.

**Figure 5 viruses-11-01057-f005:**
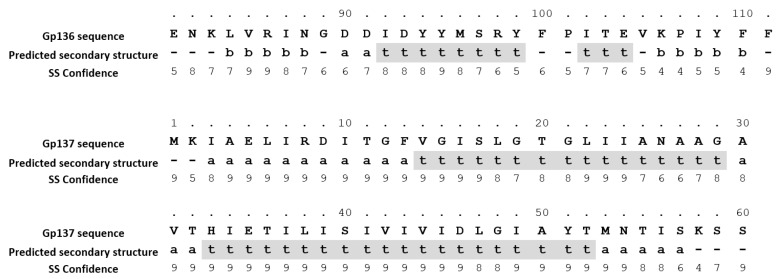
Predicted transmembrane helices for Gp136 and Gp137. a–Alpha helix, b–beta strand, t–TM helix. Numbers 0–9 represent secondary structure prediction confidence by Phyre2.

**Table 1 viruses-11-01057-t001:** Bacterial strains, bacteriophages and plasmids used in the current study.

Name	Description	Source
***E. coli* strains**		
DH10B	Used for HPUFs screening	NEB, USA
DH5α	Used for pBAD30 cloning	NEB, USA
**Bacteriophages**		
ϕR1-RT	Used as a template for amplifying all the ϕR1-RT encoded genes used in the current study	[11]
T4	Used as a template for amplifying *regB* and *ndd* control genes	[14]
**Plasmids**		
pETSmt3- DCD-1L	Used as template to amplify the *dcd-1l* control gene	[15]
pU11L4	This plasmid consists of pUC19 with a *Kpn*I-*Pst*I linker in the MCS and *luxAB* genes under expression of *ompF* promoter of YeO:3 at the *Sap*I site (Appendix A.	This study
ϕX174 RF1	Used as template to amplify the *lysE* control gene	Thermo Fisher Scientific, USA
pBAD30	Plasmid with arabinose-inducible promoter to express ϕR1-RT HPUFs in *E. coli* for confirmatory assay	[16]

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
