# Peer review of "A Toxicity Screening Approach to Identify Bacteriophage-Encoded Anti-Microbial Proteins"

_viruses, 2019, doi:10.3390/v11111057_

Round 1
Reviewer 1 Report
The manuscript describes a highly interesting approach toward identifying genes that are toxic to bacteria from the large pool of proteins without designated function found in phage genomes.
The approach is original and promising even if this procedure has some inherent imperfections which the authors discuss. However, I think the discussion should include other reasons why toxic genes could be missed using this approach. 1) Proteins could require other phage encoded proteins for proper folding or other form of posttranslational modification. 2) Some proteins also work in conjunction with others. While such multiprotein complexes are less likely to be useful for interventions the current approach will not detect such functionalities.
While I believe the method to be promising the article mentions a fact that sheds doubt on its reliability. Endolysins can be cloned without immediate detrimental effect to the cells. Only if the cytoplasmic membrane is damaged can the endolysin access the cell-wall and degrade it. If the authors freeze a culture containing the construct (without glycerol) and subsequently raise the temperature to 37 degrees C, lysis should be observed and the solution become extremely viscous as a result of DNA release (That is – if the enzymatically active domain of the protein can degrade E. coli cell walls in the first place).
However, it should not be possible to clone a holin with a promoter that is not extremely tight. These molecules are lethal and generally work across species and genera. I find it extremely difficult to believe that a Yersinia holin will not form lesions in an E. coli membrane. Perhaps an error was introduced during PCR? Perhaps part of the gene was not included in the construct? The authors should provide an adequate explanation as to why this construct did not register as a toxic gene. I feel there should be a relatively simple explanation for the observed result.
Here follow a few comments on the form of the manuscript.
At several points throughout the manuscript sentences lack articles: i.e. lines 128 “the” rest of the PCR… 157 using colony counter should be “a” colony counter, line 159 of ‘the” non-toxic control gene. The reverse is also observed. I.e. Line 111 The GenBank should be “GenBank without” an article. I suggest to let a native speaker check the entire document.
Line 377 should read: The reason “for” variation.
Line 394 It is “thought”?
The names of bacteria in the references should be in italics.
Author Response
The manuscript describes a highly interesting approach toward identifying genes that are toxic to bacteria from the large pool of proteins without designated function found in phage genomes.
The approach is original and promising even if this procedure has some inherent imperfections which the authors discuss. However, I think the discussion should include other reasons why toxic genes could be missed using this approach. Proteins could require other phage encoded proteins for proper folding or other form of posttranslational modification. Some proteins also work in conjunction with others. While such multiprotein complexes are less likely to be useful for interventions the current approach will not detect such functionalities.These comments are valid and we have now discussed this (L381-383) as follows:
In our screening approach we will not detect these interaction-dependent toxic phage products including proteins that require other phage encoded proteins for proper folding or for other form of posttranslational modification.
While I believe the method to be promising the article mentions a fact that sheds doubt on its reliability. Endolysins can be cloned without immediate detrimental effect to the cells. Only if the cytoplasmic membrane is damaged can the endolysin access the cell-wall and degrade it. If the authors freeze a culture containing the construct (without glycerol) and subsequently raise the temperature to 37 degrees C, lysis should be observed and the solution become extremely viscous as a result of DNA release (That is – if the enzymatically active domain of the protein can degrade E. coli cell walls in the first place). However, it should not be possible to clone a holin with a promoter that is not extremely tight. These molecules are lethal and generally work across species and genera. I find it extremely difficult to believe that a Yersinia holin will not form lesions in an E. coli membrane. Perhaps an error was introduced during PCR? Perhaps part of the gene was not included in the construct? The authors should provide an adequate explanation as to why this construct did not register as a toxic gene. I feel there should be a relatively simple explanation for the observed result.This is a good point, and we have payed attention to it in the revised manuscript (see lines 368-383). The following new text was added:
The genes encoding endolysin (g122) and holin (g253) were previously identified from the Ï•R1-RT genome [11]. Also gene g252 product showed similarity to a holin. As the endolysins and holins of many phages have demonstrated antibacterial activity [3], we also tested the φR1-RT endolysin and holin encoding genes in the screening by plating assay for their toxicity towards E. coli. To our surprise we did not see the expected reduction in the plating efficiency with any of the genes (Supplementary table S7). A plausible explanation for this is that T4-like endolysins and holins are not active alone, as is the case in the closely related T4 [33]: “In the absence of either the lysozyme or the holin, lysis does not occur”. T4 holin (the gene t product) is unlike many other holins not immediately toxic but requires its own endolysin for the cell lysis to occur. Analogously, a lysis gene cassette of Burkholderia pseudomallei bacteriophage ST79 contains four genes, and minimally for lysis to occur, the expression of the peptidase and holin encoding genes is required, and only moderate lysis was observed when holin alone was expressed [34]. Similar dependence of a phage holin on cognate endolysin was observed for Streptococcus suis phage SMP, where the holin alone caused only weak but with endolysin strong lysis [35]. In our screening approach we will not detect these interaction-dependent toxic phage products.
Here follow a few comments on the form of the manuscript.
At several points throughout the manuscript sentences lack articles: i.e. lines 128 “the” rest of the PCR… 157 using colony counter should be “a” colony counter, line 159 of ‘the” non-toxic control gene. The reverse is also observed. I.e. Line 111 The GenBank should be “GenBank without” an article. I suggest to let a native speaker check the entire document.We have now payed more attention to the English language throughout the mansucript
Line 377 should read: The reason “for” variation.Added as suggested.
Line 394 It is “thought”?It now reads: A plausible explanation for this is that T4-like endolysins and holins are not active alone, as is the case in the closely related T4 [33].
The names of bacteria in the references should be in italics.All references have now been checked.
Reviewer 2 Report
The report addresses an area of topical interest, namely screening for new compounds with anti-bacterial activity that might in the future lead to new antibiotics. Overall, the report reads well, the screening is described in sufficient detail and appropriate controls are taken. The method optimized protocols avoiding the inclusion of unlikely candidates for anti-bacterial action like structural genes and DNA and RNA transaction enzymes. The end of the report is somewhat abrupt. No data on the disruption of gross bacterial processes is provided, but that might be beyond the scope of the present communication, which has a focus on the screening procedure. The reader would nevertheless like to get in the discussion at least some outlook how these antibacterial activities could lead to pharmaceutical activities since the currently identified toxic gene products need to be expressed from the inside of the transformed cell, which is not suitable for practical application. Isn't a screening step missing where the supernatant of the transformed E. coli cell is tested for inhibitory activity against Yersinia enterocolitica, when added to the growth medium? Pro and contras for the current screening system should be discussed compared to published papers of the field and the objective needs for any future industrial application.
Author Response
The report addresses an area of topical interest, namely screening for new compounds with anti-bacterial activity that might in the future lead to new antibiotics. Overall, the report reads well, the screening is described in sufficient detail and appropriate controls are taken. The method optimized protocols avoiding the inclusion of unlikely candidates for anti-bacterial action like structural genes and DNA and RNA transaction enzymes. The end of the report is somewhat abrupt. No data on the disruption of gross bacterial processes is provided, but that might be beyond the scope of the present communication, which has a focus on the screening procedure.
The reader would nevertheless like to get in the discussion at least some outlook how these antibacterial activities could lead to pharmaceutical activities since the currently identified toxic gene products need to be expressed from the inside of the transformed cell, which is not suitable for practical application.We have added the following text in the discussion on this topic (lines 406-415).
The toxic phage proteins are not likely to act as such to kill bacteria extracellularly as they cannot penetrate the cell walls and membranes. Therefore, investigations on the molecular level interactions between the toxic hits and the host bacteria are required to expose their targets, which in turn could be used to screen small molecule libraries for inhibitors. We can think of two possible approaches to identify the targets. First, the phage toxin should be purified to be used as a bait in fishing out the target. As the expression of the protein in E. coli, due to its toxicity, may be difficult, yeast or plant cells could be used instead, or ultimately, in vitro translation could be applied. Second, genetic approach where spontaneous toxin-insensitive mutants would be isolated and the target identification would be carried out by whole genome sequencing of the mutants.
Isn't a screening step missing where the supernatant of the transformed E. coli cell is tested for inhibitory activity against Yersinia enterocolitica, when added to the growth medium?
We have not considered this as necessary as we are targeting toxicity to intracellular targets, and, furthermore, it is very unlikely that the toxic proteins could penetrate into the bacteria.
Pro and contras for the current screening system should be discussed compared to published papers of the field and the objective needs for any future industrial application.This has been added to the discussion (lines 330-339, 393-402, and 408-415).
Reviewer 3 Report
This manuscript describes a method for identifying potentially toxic gene products from bacteriophage, focuses on hypothetical proteins of unknown function. The experiments are technically sound and the results are convincing, but I didn't feel that the method was novel enough to justify publication without the inclusion of any follow up experiments on the four identified proteins (e.g., identifying the target, confirming toxicity in natural host, etc.). Similar methods have been published, and the relatively small modifications done here don't necessarily warrant publication.
Author Response
This manuscript describes a method for identifying potentially toxic gene products from bacteriophage, focuses on hypothetical proteins of unknown function. The experiments are technically sound and the results are convincing, but I didn't feel that the method was novel enough to justify publication without the inclusion of any follow up experiments on the four identified proteins (e.g., identifying the target, confirming toxicity in natural host, etc.). Similar methods have been published, and the relatively small modifications done here don't necessarily warrant publication.
Thank you for your critical comments on our manuscript “Identification of four antibacterial peptides by toxicity screening of bacteriophage Ï•R1-RT encoded hypothetical phage proteins”.
Our aim for this manuscript was to create a systematic, efficient and high-throughput screening method to find potential toxic phage proteins among those annotated as hypothetical proteins of unknown function (HPUFs). The emphasis of our study was on the very early stage of the whole lineage of the studies on phage HPUFs, rather than investigating their molecular functions. Of course the mechanism of toxicity will be studied further. We feel that each successfully identified target will generate a publication on its own and therefore we think that it is outside the scope of the present manuscript.
Indeed, several research groups have also been searching for toxic phage proteins and developed different ways for the screening but not as systematically as we. Up to date, many have focused on individual phages, i.e. the researches have in-depth investigated a phage from identification of toxic hypothetical proteins to functional elucidation of the proteins. The existing criteria for selecting candidate HPUFs to screen at the first place are mainly based on literature. Yet, in our study, we had experimental data from LC-MS/MS to filter out the falsely-annotated-as-hypothetical by truly-to-be-structural proteins as the first step.
Moreover, our study has the potential to be applied to a broad range of phages for toxic HPUFs, as the LC-MS/MS and cloning-based screening and validation systems are not restrained to a single strain/species. In fact, in our lab, we have used the same method to screen a Klebsiella phage. Like the findings in the current study, approximately 10 % of HPUFs from Klebsiella phage were found toxic (unpublished results).
Surely, our method can be biased due to the selection of the indicator strain E. coli. Nevertheless, we are looking for more general targets of toxic phage proteins, which possibly lead to further investigations on small molecule inhibitors. Our method is robust, and to our knowledge, this could be the first screening method to increase the volume and speed of screening toxic proteins from broader range of bacteriophages.
Round 2
Reviewer 3 Report
The readability of the revised manuscript is much improved. I still have some reservations on the novelity of the experiments, but the authors have made a good case that the manuscript represents a useful guide for identifying toxic phage proteins. I support publication in its current form.